# Research on a Blue–Green LED Communication System Based on an Underwater Mobile Robot

**Tianhao Shen [1], Junfang Guo [1,\*], Hexi Liang [2], Yanlong Li [3], Kaiwen Li [3], Yonghong Dai [4] and Yong Ai [4,5]**

1 School of Information Engineering, Wuhan Huaxia Institute of Technology, Wuhan 430223, China; shenth@whhxit.edu.cn

2 School of Computer and Information Engineering, Hubei Normal University, Huangshi 435002, China; hexiliang@whu.edu.cn

3 School of Information and Communication, Guilin University of Electronic Technology, Guilin 541004, China; lylong@guet.edu.cn (Y.L.); xchen@guet.edu.cn (K.L.)

4 School of Electronic Information, Wuhan University, Wuhan 430072, China; yhdai@whu.edu.cn (Y.D.); ay@whu.edu.cn (Y.A.)

5 Optical Communicaion Equipment Research and Development Department Wuhan Liubo Optoelectronic Technology Co., Ltd., Wuhan 430072, China

\* Correspondence: guojf@whhxit.edu.cn

**Abstract:** Underwater robots have been widely used in ocean exploration, deep-sea observation, seabed operations, marine scientific research, and other fields. Underwater low-latency, efficient, and safe communication modes are key to realizing the application of an underwater robot data transmission system. This paper mainly studies the optical communication between underwater mobile robots, including the large-dispersion-angle light-emitting diode (LED) design, large field of view receiving technology, weak light detector technology, etc. By designing a 120° large divergence angle underwater optical communication system in this study, the receiving field-of-view angle of the receiving end can reach 60°, which is suitable for the optical communication system of an underwater mobile platform. The high-power LED driver circuit is designed to drive the high-power LED and adopt weak light detection technology to ensure its stability and reliability. The experimental results show that, in the case of incomplete alignment between the transmitter and receiver, stable communication of underwater robots in motion is achieved through the design of a large divergence angle and a receiving field-of-view angle and the use of an underwater weak light detection technology. The communication distance is 30 m, and the communication rate remains above 10 Mbps. The information transmission content can include network data transmission, real-time video, high-definition video, high-definition images, and other data types. This equipment provides a solution for cableless data transmission of remotely operated vehicles (ROVs) and substantially enhances the application field of ROVs.

**Keywords:** underwater optical communication; LED; ROV; data transmission

## 1. Introduction

With the arrival of the 6G era, the integrated communication network layout of space–air–ground–sea is gradually taking shape. Human exploration of the underwater environment is becoming increasingly frequent, therefore the requirements for the transmission rate, transmission distance, and communication quality of underwater wireless communication technology are also increasing [1].

Currently, underwater exploration and exploitation rely on underwater robots, and underwater robotic communication is the key technology for underwater exploitation. Underwater wireless optical communication (UWOC) is an emerging high-speed underwater communication technology [2]. Compared with the traditional underwater electromagnetic wave communication and underwater acoustic wave communication technology, UWOC

has the advantages of high transmission bandwidth, strong resistance to electromagnetic interference, low power consumption, and small size. As a result, it is gradually attracting widespread attention in both academia and industry. Despite the large attenuation of light as an information carrier in underwater transmission, its transmission distance can still reach up to a level of 100 m [3]. This is sufficient to meet the distance requirement of sensor data transmission, so it is favored by the research community due to the aforementioned appealing potential. In order to achieve reliable long-distance communication and high-speed data transmission links, light sources with high emitting power and photodetectors with high sensitivity are usually used to extend the communication distance [4]. At the same time, advanced signal processing techniques are used to increase the communication rate.

In recent years, with the development of communication technology, various research groups across the globe have gradually realized a UWOC at a 100 m level and a high-speed transmission over a short distance Gbit/s level. In [5], 85 m/80 kbit/s communication was achieved in a laboratory sink using a 532 nm laser and a single-photon counter. Based on this, a 120 m transmission experiment was demonstrated in the following year [6]. In [7], a UWOC system with a transmission distance of 34.5 m and a transmission rate of 2.7 Gbit/s was constructed using a 20 mW laser and a binary on–off keying (OOK) signal. In [8], 55 m/6.6 Gbit/s transmission was achieved using a Discrete Multitone (DMT) modulation technique and a nonlinear equalizer. In [9], the Non-Return-to-Zero On–Off Keying (NRZ-OOK) modulation technique was used to achieve 100 m/500 Mbit/s communication in a tap channel. This takes into account the advantages of the high transmission rate of the UWOC system and the need for transmission distance. Subsequently, Jing Xu's group at Zhejiang University successively achieved a 150 m and 200 m distance and a 500 Mbit/s data transmission in a 50 m pool [10]. After further optimization of the equalizer complexity in [11], a wide-bandwidth photomultiplier tube (PMT)-based transmission at the 100 m/1 Gbps level was achieved. Underwater transmission of 40 m/1 Gbps was achieved in [12] using a high-bandwidth single-photon counter. So far, UWOC has opened the era of a 100 m high-speed link. In practical applications, due to the complexity of the underwater environmental factors, such as underwater planktonic microorganisms, underwater turbulence and other interference factors will have certain impacts on underwater optical communication. Particularly, mobile underwater communication systems need to consider the design of underwater optical systems, such as the transmitting angle, receiving field-of-view angle, and other key factors [13,14].

LEDs have the advantages of low cost, long lifetime, and high optical-to-electrical conversion efficiency. However, the light emitted by LEDs has poor coherence, large linewidth, low bandwidth, and a large beam divergence angle, thus limiting the transmission distance and transmission rate of UWOC systems using LEDs as light sources [15]. To overcome these problems, an arrayed LED light source can be used to increase the transmission distance [16], and a modulation technique with higher spectral efficiency can be used to increase the transmission rate [17]. In 2009, Massachusetts Institute of Technology (MIT) researchers developed a bidirectional, high-speed UWOC system [18], which consisted of an arrayed light source and a detector of six 5W-blue LEDs. In a 30 m clear-water pool, the transmission rate reached 1.2 Mbit/s, and in a turbid pool with 3 m visibility and 9 m length, the rate was 0.6 Mbit/s. In [19], a data transmission rate of 3.4 Gbit/s was achieved using an array light source composed of six LEDs in a clear-water channel with a length of 4.5 m. However, due to the directional propagation characteristics of the LED beam, a small divergence angle made the LED beam coverage limited; the LED divergence angle was too large and led to a decline in beam intensity, which affected the quality of communication. Especially in the process of moving, an underwater communication system needs to consider the underwater optical system itself, designed to transmit the angle, receive the field-of-view angle, and other factors. At the same time, it is also necessary to consider the problem of real-time communication in the mobile state of underwater robots.

In order to achieve wireless data transmission for mobile underwater robots, an underwater optical communication system with a 120° large dispersion angle is designed

in this work. By adopting high-power LED modulation technology, the optical power of an LED can reach more than 5W when the modulation rate is 10 Mbps. In addition, considering the high underwater alignment and the influence of underwater turbulence on the underwater robot in a moving state, the system adopts a photomultiplier tube with a large photosensitive surface and high sensitivity as the photodetector. The sensitivity of the detector reaches more than −55 dBm, and the field of view reaches more than 60°. The system adopts a large dispersion angle and large field-of-view receiving angle technology for underwater robots in mobile states to maintain stable communication using high-power LED emission and high-sensitivity detection technology.

## 2. The Blue–Green LED Communication System of an Underwater Mobile Robot

The proposed underwater optical communication system uses LEDs of 450 nm and 520 nm as optical carriers to achieve bidirectional data transmission. The system is mainly composed of an LED and its driving circuit, a large-area photomultiplier tube and its photoelectric conversion circuit, a field-programmable gate array (FPGA) for signal processing, and computer control software, as shown in Figure 1. The main functional part of FPGA is the information processing unit. The underwater robot as a data source interacts with the FPGA through Ethernet, and the data source sending and data stream receiving are accomplished through network communication. The FPGA processes the data using 8 B/10 B encoding and parallel–serial conversion. The processed data are inputted as a low-voltage transistor–transistor logic (LVTTL) level signal to the LED driver circuit, which loads the data onto the light signal. The light signal is transmitted to the receiving end through the underwater channel after adjusting the divergence angle via the transmitting antenna. The optical signal is converged by the receiving optical antenna onto the photosensitive surface of the large-area PMT to be converted into photocurrent, which is converted into a voltage signal by the transimpedance amplifier (TIA) circuit. Then, subsequent amplification and level conversion are performed, and finally, the LVTTL level is provided as an input to the FPGA. After that, the FPGA performs serial-to-parallel conversion and 8 B/10 B decoding to recover the data, and then returns the latter data to the terminal at the receiving end via network communication [20,21].

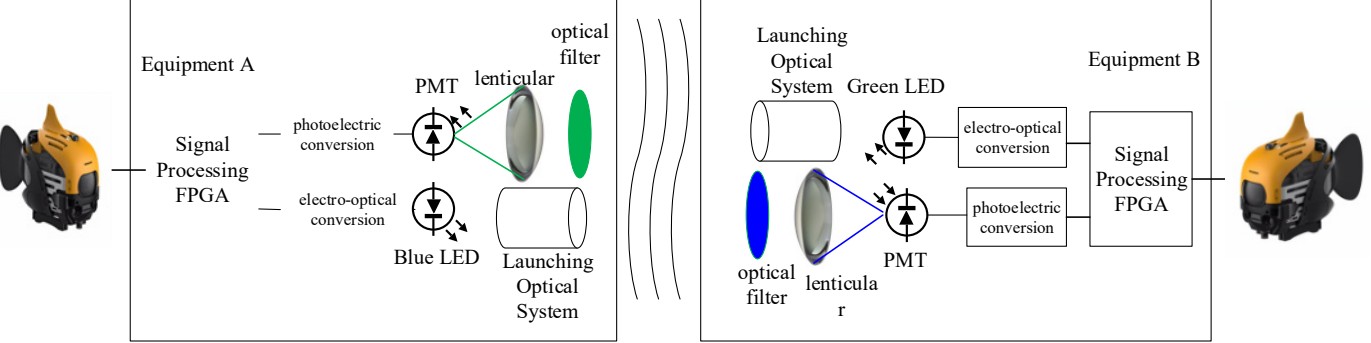

**Figure 1.** The design scheme of blue–green LED communication system of an underwater mobile robot.

## 3. Optical Design

### 3.1. Underwater Optical Communication Link Analysis

The underwater mobile optical communication system needs to be set before the design of the initial parameters, such as the transmission of optical power, light source emission angle, detector receiving sensitivity, and several other indicators [22]. According to these indicators and various factors on the optical transmission loss caused by the link analysis, we set the design parameters of the underwater optical communication system as shown in Table 1.

**Table 1.** Performance parameters of underwater optical communication system.

| Parameter Name | Parameter Value |
|---|---|
| Light source type | LED |
| Emitted optical power ($P_T$) | 5W (37 dBm) |
| Caliber of launch ($\gamma$) | 8 cm |
| Angle of divergence ($\theta$) | 120° |
| Receiving caliber ($\phi$) | 30 mm |
| Receiver sensitivity ($P_R$) | $-55$ dBm |
| Communication distance (d) | 30 m |
| Water quality losses (WL) | 0.5 dBm/m |

Light in underwater transmission is mainly divided into loss $P_{EL}$ at the transmitting end, loss $P_{RL}$ at the receiving end, loss $P_{GL}$ formed by geometric expansion, and loss $P_{WL}$ caused by underwater attenuation. Among these losses, two kinds of loss $P_{EL}$, $P_{RL}$ are caused by optical components, while two kinds of loss, $P_{GL}$ and $P_{WL}$, are mainly related to distance. The link margin of the underwater optical communication system can be calculated using the following equation [23]:

$$P_{Lm} = P_T - P_{EL} - P_{RL} - P_{GL} - P_{WL} + P_R. \tag{1}$$

As shown in Equation (1), $P_T$ represents the total transmitted optical power and $P_R$ is the received sensitivity. Generally, in the underwater optical communication system, the optical loss $P_{EL}$ at the transmitting end is about 1.5 dB, and the optical loss $P_{RL}$ at the receiving end is about 2 dB. From the data in Table 1, it can be concluded that the underwater attenuation loss $P_{WL}$ is about 15 dB.

Underwater robots are prone to the emergence of the state of transmitter off-axis receiver horizontal angle $\alpha$ and transmitter horizontal receiver off-axis $\beta$ in the process of mobile optical communication, as shown in Figure 2.

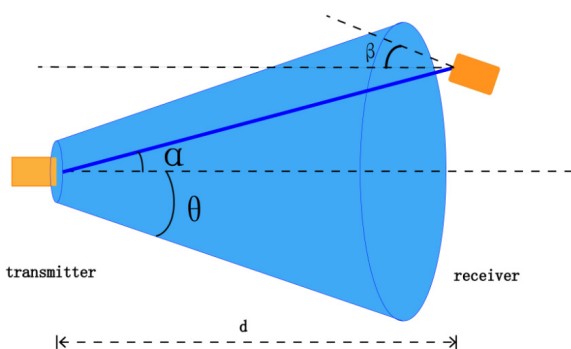

**Figure 2.** Link analysis model of underwater mobile optical communication.

Thus, the geometric loss can be calculated using Equation (2):

$$P_{GL} = \frac{\left(\frac{\gamma}{2}\right)^2 \times \cos(\alpha + \beta)}{(d \times \tan\theta)^2}. \tag{2}$$

Combining the attenuation analysis data, the following expression for the link margin $P_{LM}$ can be calculated:

$$P_{LM} = 73.5 - 10 \times \log\left[\frac{\left(\frac{\gamma}{2}\right) \times \cos(\alpha + \beta)}{(d \times \tan\theta)}\right]. \tag{3}$$

According to the above design parameters, we analyze the relationship between the transmitter off-axis receiver horizontal angle $\alpha$ and transmitter horizontal receiver off-axis

β, as shown in Figure 3. The actual environment of underwater optical communication is more complex and generally requires link redundancy greater than 5 dBm or more as a guarantee for basic communication [24]. According to the curve, it can be seen that when α = 60° and β ≈ 0° ∼ 10°, underwater optical communication can be established normally. When β > 10°, the angle of α needs to be adjusted to ensure the stability of underwater optical communication.

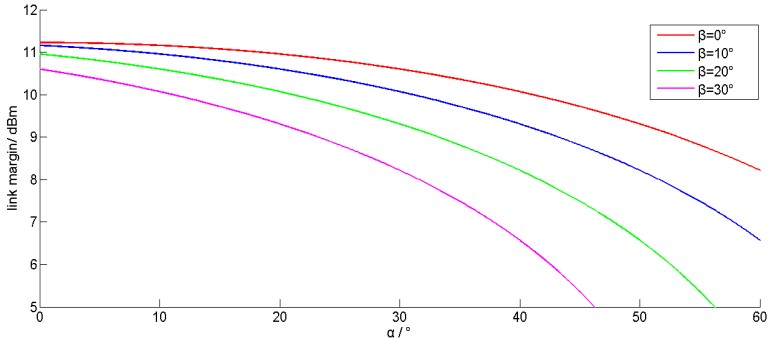

**Figure 3.** Relationship between link redundancy and device deflection angle.

In the actual test, we used the method of collecting communication water samples several times to measure the degree of light attenuation in the water body. We utilized an OSRAM PLT5 520B laser to pass through 1 m of water in FRP, respectively. A PM100D optical power meter was used to receive the laser light passing through the fiberglass, and the degree of light attenuation in this water quality was calculated and compared to the degree of light attenuation in free space. The measurement data are shown in Table 2.

**Table 2.** Data measurements of water quality attenuation levels.

| Number of Experiments | Laser Output | Received Optical Power through Water | Received Optical Power through Free Space | Loss Value |
|---|---|---|---|---|
| 1 | 6.5 dBm | 4.18 dBm | 4.98 dBm | 0.8 dB/m |
| 2 | 6.5 dBm | 4.20 dBm | 5.00 dBm | 0.8 dB/m |
| 3 | 6.5 dBm | 4.21 dBm | 5.01 dBm | 0.8 dB/m |
| Average value | 6.5 dBm | 4.20 dBm | 5.00 dBm | 0.8 dB/m |

From the data in the table, it can be seen that the average value of the received optical power passing through the free space is 5 dBm and the average value of the received optical power passing through the water is 4.20 dBm, and the water quality loss was measured to be about 0.8 dB/m.

### 3.2. Transmitting Optical Design

The light source on the transmitting side was chosen to be Lattice 45 mil 450 nm LED and SA 38 mil 525 nm LED, produced by Hangzhou Yuanfang Optoelectronic Information Co. (Hangzhou, China). The optical power of the two single LEDs was 500 mW and the emission angle was 120° [25]. Lighttools software was used to simulate the layout of 10 LEDs, with the simulation results shown in Figure 4. Figure 4a gives the simulation of the light tracing of 10 LEDs, and Figure 4b shows the dispersion angle and light intensity distribution of the LED on a circular angle scale. The green curves in Figure 4b represent the energy intensities corresponding to different divergence angles. From the figure, it can be seen that the actual dispersion angle is greater than 120°, which meets the requirements of the optical design of the transmitter.

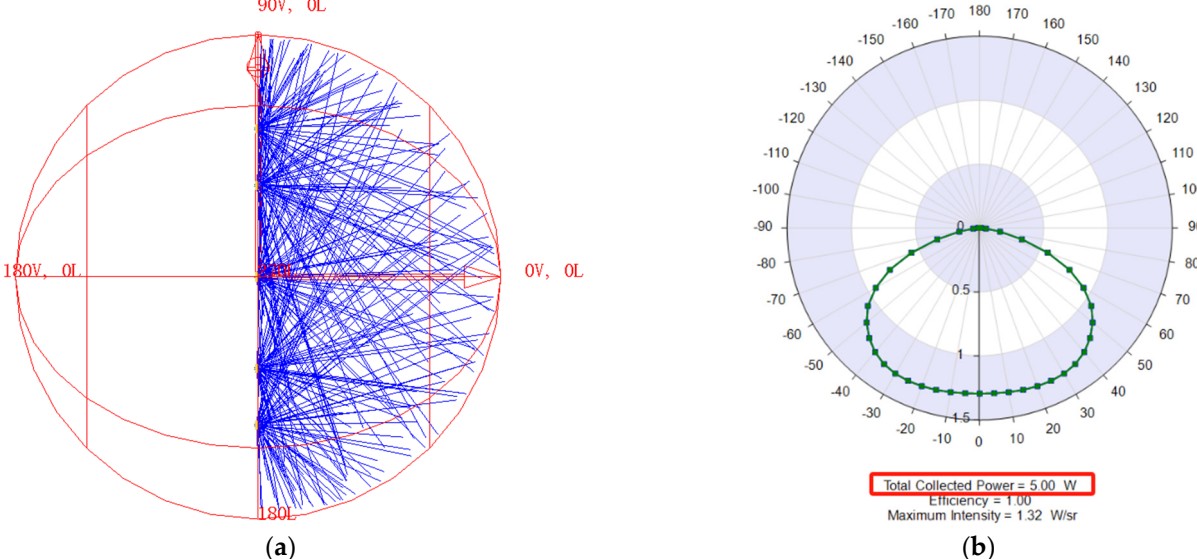

**Figure 4.** Transmitting optical intensity simulation. (**a**) Light tracing simulation diagram of LED array. (**b**) Light intensity distribution map.

### 3.3. Receiving Optical Design

In the field of underwater robotic communication, in order to ensure the stable transmission of optical communication systems, two methods are usually used: using an automatic tracking system or using a detector with a large receiving field of view angle [25]. An automatic tracking system uses a high-frequency Complementary Metal-Oxide-Semiconductor (CMOS) camera to capture and analyze the optical communication spot and extracts the coordinates to control the precision torque motor or precision gimbal to rotate the receiving optical system and precisely capture the optical signals. The automatic aiming system has a large field-of-view angle of the receiving dynamic range, which can effectively ensure the stability of mobile optical communication. However, the method is costly, bulky, and difficult to implement. The large field of view receiving detector is mainly determined using the receiving lens and the photosensitive area of the receiving detector.

Let the receiving lens be an aspherical lens with an aperture of D mm and a focal length of F mm. A filter with an aperture of D mm is placed in front of the aspherical lens to filter out the interfering light and focus the signal light onto the large-area detector. Neglecting the effects of light diffraction limitation, the position of the focused spot on the detector is constantly changing as the receiving optics are constantly biased. When the spot exceeds the effective area of the detector, normal underwater optical communication transmission is not possible. The half-angle field of view of the receiving optics $\omega$ is approximately determined using the ratio of the radius R of the detector to the focal length F of the lens.

$$\omega = \arctan\frac{R}{F}. \tag{4}$$

According to the current process, the receiving end lens aperture D and focal length are positively correlated; that is, the larger the aperture, the longer the focal length, resulting in a smaller market angle. This design comprehensively considers the factors affecting the receiving field of view, and the large field-of-view angle optical receiving system is realized according to the link simulation budget and optical simulation. In the receiving end, the Hamamatsu H14447 photomultiplier was chosen as the photodetector, with a 25 mm photosensitive area and a 21 mm receiving lens. The half-angle field of view was calculated to be 30.7°, and the full-angle field of view was 61.4° according to Equations (2)–(4). According to the design parameters, it was simulated and analyzed as shown in Figure 5a.

Figure 5b gives the distribution of light energy at a half angle of 31°. The average energy was about −80.64 dBm, which was in accordance with the results of the theoretical analysis provided above.

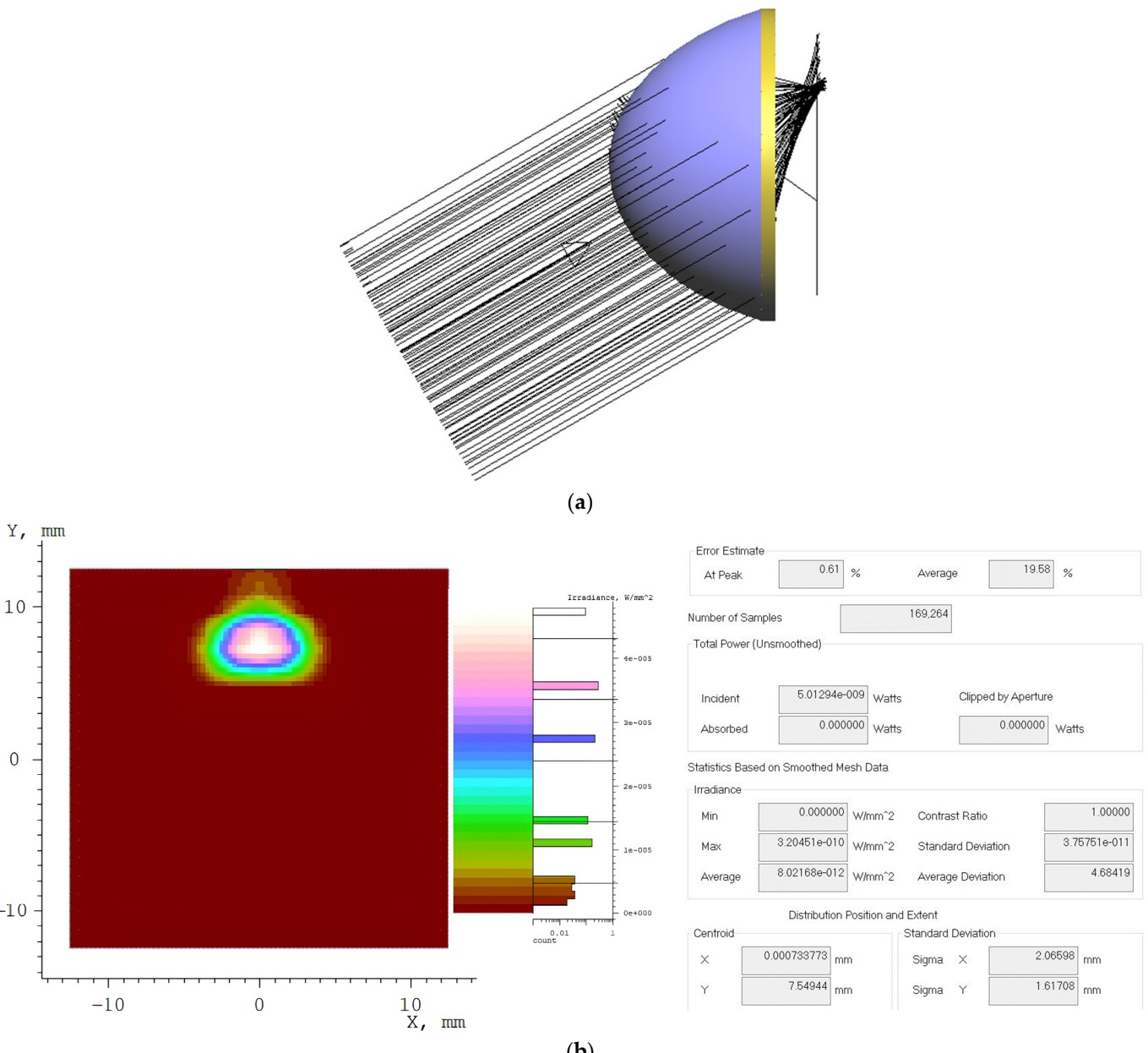

**(a)**

**(b)**

**Figure 5.** Receiving optical simulation analysis. (**a**) Receiving optical simulation schematic for a half angle of 31°. (**b**) Optical energy distribution at a half angle of 31°.

## 4. System Hardware Circuit Design

Considering the complexity of the underwater environment, to ensure that the transmission distance of the underwater optical communication system is long, OOK modulation technology is used at the transmitting side. Together with the Bias-T circuit model, it achieves an LED transmitting optical power up to 5 W, while the modulation rate of the LED reaches up to 10 Mbps. The diagram displaying the modulation technology for high-power LEDs can be observed in Figure 6. High-power LED modulation can be established due to the level conversion, gallium nitride (GAN) drive, high-speed switching, DC bias, feedback control, and monitoring of five modules.

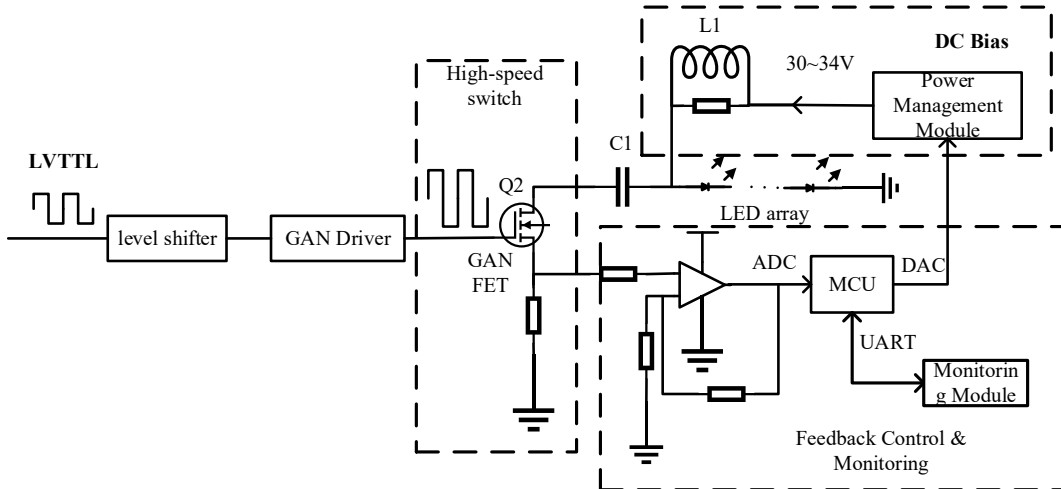

**Figure 6.** Principle block diagram of high-power LED modulation technology.

The key element of high-speed modulation design for the transmitter section of the underwater optical communication system lies mainly in the selection of high-speed switches. Most of the current research selects conventional high-power N-Metal-Oxide-Semiconductor (NMOS) tubes as high-speed switches. This design uses a gallium nitride field effect transistor (GAN FET) as a high-speed switch. It has the advantages of nanosecond switching frequency, high-power density, high conversion efficiency, low energy loss, and a tens-of-megabits modulation rate with a high-speed GAN driver. In order to adjust emitted light power and to adapt to multiple LED arrays, this design incorporates a feedback control circuit. Using the C8051F007 microcontroller unit (MCU) to monitor the current flowing through the LED array and form feedback, the output voltage of the power management module is controlled to achieve its function. At the same time, the MCU can monitor the abnormal condition of the transmitter circuit. If the circuit is in an abnormal state, the MCU module sends abnormal information to the monitoring module via the Universal Asynchronous Receiver/Transmitter) (UART) for easy maintenance.

The receiving end of the system uses a PMT as a photodetector, which has the characteristics of a large photosensitive area and high sensitivity. If the light intensity is too high, it is easy for the PMT to become saturated, resulting in the PMT losing its ability to work. If the PMT is saturated for a long time, it will be permanently damaged and become nonfunctional. The optical signals are converted by the PMT and TIA and amplified by an operational amplifier (OPA). The signal is converted to the LVTTL level by level conversion and input to the signal processing platform FPGA. Figure 7 presents the design of the PMT signal processing circuit.

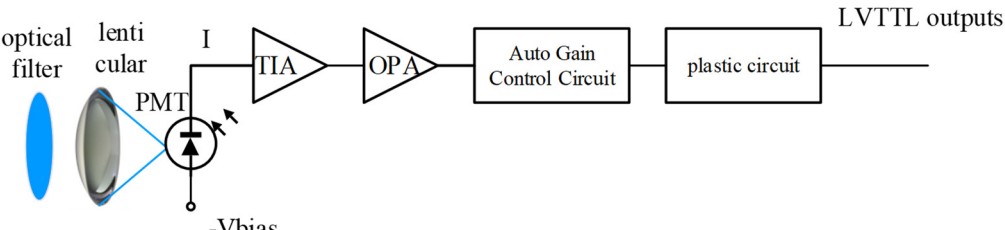

**Figure 7.** Signal processing circuit of PMT.

As the optical signal travels through the underwater channel towards the receiver, it first passes through the filter to filter out the rest of the clutter [26]. It then passes through the optical antenna and converges through the optical antenna onto the photosensitive surface of the large-area PMT. At this time, the voltage signal amplitude is usually very small, and low-noise operational amplifiers are used to achieve low-amplitude voltage

signal amplification. Due to the movement of underwater robots, the communication distance and communication angle may change at any time, which continuously changes the received amplitude level. The inconsistency of the PMT response at 450 nm and 520 nm is reflected in the inconsistency of the amplitude of the TIA output. To ensure the stability of the communication link, this design adopts an automatic gain control circuit using a VCA820IDGSR model. It can detect the amplitude of the input information and adaptively adjust the amplitude level. After testing, without adding an automatic gain control circuit, the dynamic range of the optical signal received by the optical communication is about 20 dBm. After adding the automatic gain control circuit, the dynamic range of the signal received at the receiving end is increased to 30 dBm. Finally, a comparator is added as a shaping circuit to output an LVTTL-level signal.

According to the aforementioned design scheme for the transmitter and receiver, we created the corresponding circuit schematic and printed circuit board (PCB) to authenticate the feasibility of the scheme. We used a signal generator to transmit nonsequential data to test the modulation rate of the transmitter circuits at 5 Mbps and 10 Mbps, and we measured the BER of the signals to be 0 at 5 Mbps and 10 Mbps. Subsequently, we observed the eye diagram of the received signal, as shown in Figure 8. As shown in the figure, the signal quality at a modulation rate of 5 Mbps is better than that at a modulation rate of 10 Mbps. However, a modulation rate of 10 Mbps can still achieve good communication quality.

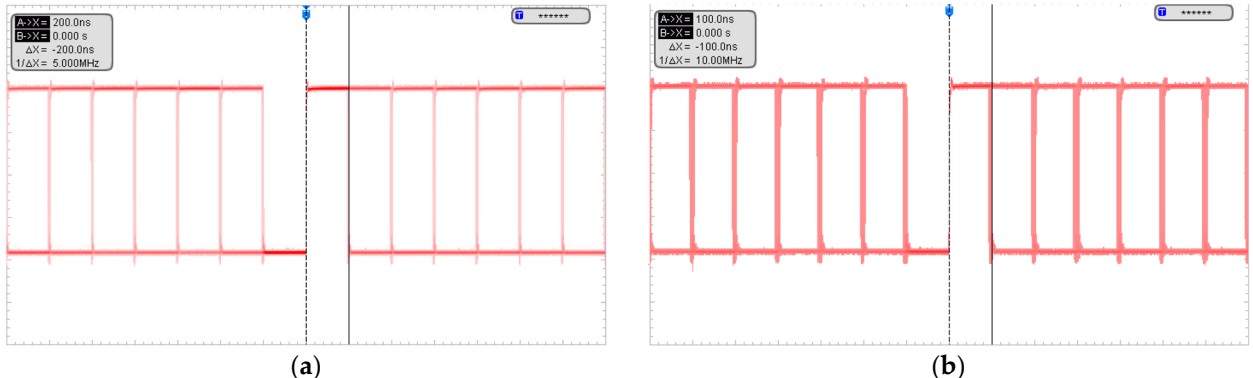

(a)                                            (b)

**Figure 8.** Eye diagram. (**a**) Eye diagram of the 5 Mbps transmission date. (**b**) Eye diagram of the 10 Mbps transmission date.

## 5. Underwater Optical Communication Experiment

In this section, we will verify the performance of the designed underwater mobile robot through experiments. The optical communication prototype structure of the underwater mobile robot was designed based on our block diagram and it can be seen in Figure 9.

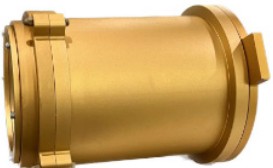

**Figure 9.** Engineering prototype.

The underwater experiment was carried out in a swimming pool with a length of 50 m and a width of 25 m. The underwater attenuation of the swimming pool was tested before the start of the experimental demonstration, and the average of several tests was taken to make the underwater attenuation of the swimming pool about 0.8 dBm/m. As shown in Figure 10, the two engineering prototypes are installed on two underwater robots respectively, the underwater robots were used to keep transmitting information in motion and observe whether they could communicate properly.

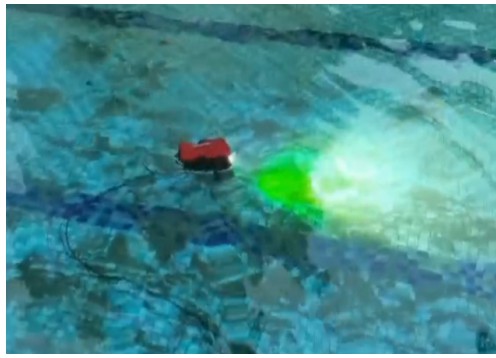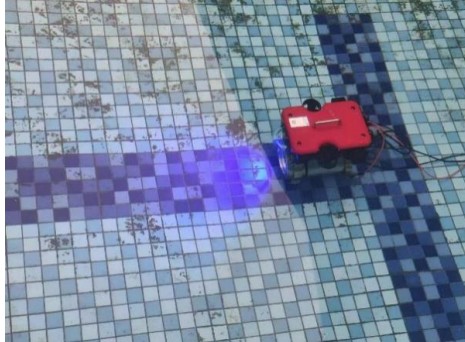

**Figure 10.** Underwater robots working with optical communication.

To make it easier to analyze the performance index between the underwater optical communication and the underwater robot, one underwater robot was kept relatively stationary, as shown in Figure 11. We observed whether the other underwater robot was communicating normally through constantly adjusting the communication angle and sending real-time video, bit error rate (BER), and other information.

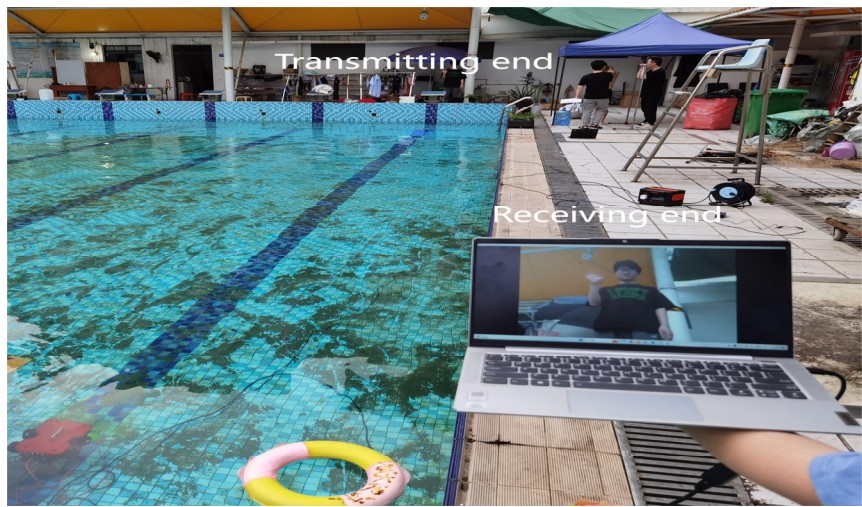

**Figure 11.** Underwater robot communication diagram.

Several experimental trials were conducted to test the optical communication between the underwater robots. When the relative deflection angle between the transmitter and receiver was 0°, the communication distance was up to about 30 m. When the transmitter and receiver were biased at a larger angle, the communication distance was more affected. The test data are shown in Table 3.

**Table 3.** Transmitter and receiver bias test.

| Communication Distance | 0° Deflection of the Transmitting Side of the Device | 15° Deflection of the Transmitting End of the Device | 30° Deflection of the Transmitting End of the Device |
|---|---|---|---|
| | Real-time video/BER | Real-time video/BER | Real-time video/BER |
| 15 m | Normal/0 | Normal/0 | Normal/0 |
| 20 m | Normal/0 | Normal/0 | Normal/0 |
| 25 m | Normal/0 | Normal/0 | Normal/0 |
| 30 m | Normal/0 | Normal/0 | Normal/0 |
| 33 m | Screen Stuttering/$10^{-7}$ | Loss of screen/$10^{-6}$ | No screen/$10^{-5}$ |

According to the analysis model for underwater optical communication links, the deflection angle would benefit from being larger. The model was based on a reference

water quality of 0.5 dBm/m, whereas the actual test water quality was 0.8 dBm/m. This discrepancy was caused by inconsistent attenuation levels in the water quality, resulting in a degree of error between the theoretical model and the actual test data.

Our experiments validate that underwater robots with underwater optical communication systems can achieve high-speed wireless information transmission, and the transmission distance has a greater relationship with the deflection angle of the device. According to the test results, when the transmitting and receiving devices are deflected to $0°$, the communication BER is $10^{-7}$ when the communication distance is 33 m. When the transmitting device is deflected to $15°$, the BER is $10^{-6}$ at the same communication distance, and when the transmitting device is deflected to $30°$, the BER is observed as $10^{-5}$ for the same communication distance.

## 6. Conclusions

As a key technology for high-speed underwater data transmission, underwater optical communication is highly suitable for establishing high-speed communication between underwater mobile robots. In order to improve the coverage of the beam, this study designs a $120°$ large divergence angle underwater optical communication system where the receiving field-of-view angle of the receiving end can reach $60°$, which is suitable for the optical communication system of the underwater mobile platform. A high-power LED driver circuit is designed to drive the high-power LEDs, and weak light detection technology is used to ensure the stability and reliability of the communication system. The experimental results show that, in the case of incomplete alignment between the transmitter and receiver, the stable communication of underwater robots in motion is achieved through the design of a large divergence angle and receiving field-of-view angle and the use of underwater weak light detection technology. The communication distance is 30 m, and the communication rate remains above 10 Mbps. When the transmitter is deflected to $30°$, the BER for 30 m of communication is $10^{-5}$. This study lays the foundation and reference value for the subsequent development of communication in underwater mobile platforms.

**Author Contributions:** Conceptualization, T.S. and J.G.; methodology, T.S. and H.L.; formal analysis, T.S., J.G. and H.L.; investigation, T.S. and J.G.; resources, Y.L. and Y.D.; data curation, Y.L., J.G. and Y.A.; writing—original draft preparation, T.S., J.G., Y.L. and K.L.; writing—review and editing, T.S., J.G., Y.D. and Y.A.; visualization, Y.A. and T.S.; supervision, Y.D.; project administration, J.G.; funding acquisition, Y.D. All authors have read and agreed to the published version of the manuscript.

**Funding:** National Key R&D Program of China (2022YFB3903800), Hubei Natural Science Foundation of China (2022CFD045), Hubei Province Science and Technology Department Project of China (2021BAB099), and Wuhan Huaxia Institute of Technology Research Fund Key Project (22007).

**Institutional Review Board Statement:** Not applicable.

**Informed Consent Statement:** Not applicable.

**Data Availability Statement:** Data underlying the results presented in this paper are not publicly available at this time but may be obtained from the authors upon reasonable request.

**Conflicts of Interest:** The authors declare no conflict of interest. The authors declare that they have no known competing financial interests or personal relationships that could have appeared to influence the work reported in this paper.

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
