# Peer review of "Research on a Blue–Green LED Communication System Based on an Underwater Mobile Robot"

_photonics, doi:10.3390/photonics10111238_

Round 1

Reviewer 1 Report

Comments and Suggestions for Authors

Dear authors:

This work constructs a visible light communication system for communication between underwater vehicles using blue and green LEDs. This is a comprehensive research point, including the selection of basic devices, circuit design, packaging, code and so on. You guys really did a lot of work.

But this article has some problems:

1.The quality of the illustrations is not good enough, most look like software screenshots, and the coordinate legends are not clear enough.

2.Many of the citations are not marked in the text, which is not in line with academic norms.

3. According to the data given in Table 1 and the optical power data of a single LED in the following article, you used ten 500mW-LEDs to obtain 5W of transmission power. I think the data obtained by multiplying the number of LEDs with the optical power of a single LED is incorrect.

4. Why do you use two different colors LEDs instead of the same color LEDs? The article does not fully explain this point. The sensitivity of detector is not the same to different colors. I think if you use different colors of LEDs and the same PD as the receiving and transmitting terminals, it may lead to different communication performance. Have you ever tested the differences between these two different color links through experiments? For example, which color LED was used to measure the underwater attenuation data?

5. Can Figure 8 illustrate the performance differences of communication systems at different transmission rates? I don't see a significant performance difference. I think that when the transmission rate changes, the quality of the received signal will also change. Can you explain the difference between the two different rates in Figure 8.

6.The modulation mode used for the data in Figure 8 is not explained, including the modulation mode used by the underwater communication system finally.

4. From the perspective of communication common sense, it is difficult to avoid the bit error in the actual digital communication system. However, as can be seen from the data in Table 2, the BER from 15m to 30m is 0. Can this BER data be understood as the data after error correction? Whether the BER without error-correcting coding has been tested?

Finally, I suggest that the models and manufacturers of LED and PD devices used in this paper should be indicated, and all simulation software and methods should also be listed. The adjustment range of automatic gain control used in the system and the saturation problem of PMT can also be explained.

With best wishes

Author Response

Responses to Reviewer's Comments (Reviewer 1)
Author's Notes
Thanks for your suggestion, I have revised the article as follows:

  1. The quality of the illustrations is not good enough, most look like software screenshots, and the coordinate legends are not clear enough.

Answer: Figures have all been revised.

  1. Many of the citations are not marked in the text, which is not inline with academic norm.

Answer: Citations are labeled according to academic norm.

  1. According to the data given in Table 1 and the optical power data of a single LED in the following article, you used ten500mW-LEDs to obtain 5W of transmission power. I think the data obtained by multiplying the number of LEDs with the optical power of a single LED is incorrect.

Answer: That's a good comment. Generally the optical power of a single LED and the number of LEDs really can not be multiplied to get the total power, but the design of the 10 LEDs for the layout of the simulation and each LED is set to 500mw simulation, the simulation results for the photoelectricity of the results of the detection of the 5W, see the figure in the paper.

  1. Why do you use two different colors LEDs instead of the same color LEDs? The article does not fully explain this point. The sensitivity of detector is not the same to different colors. I think if you use different colors of LEDs and the same PD as the receiving and transmitting terminals, it may lead to different communication performance. Have you ever tested the differences between these two different color links through experiments? For example, which color LED was used to measure the underwater attenuation data?

Answer: Thank you for your question. The system realizes a bidirectional link communication where both sides can send and receive at the same time in order to achieve the full duplex function, so two different wavelengths of LEDs are used as the light source and a high cutoff filter is used to isolate the interfering light and realize its function. Different colors of light do lead to inconsistent sensitivity of the PD, in the selection of the PD is selected for the visible light PD, its sensitivity is a little gap but the degree of impact on the communication is not significant. In this paper, a green laser is used to measure underwater attenuation data. In 1963, Seibert Q and Duntley et al. found that in seawater using the wavelength of 450nm to 550nm light for transmission, the attenuation is much smaller than other bands of light. (Modified in the text)

  1. Can Figure 8 illustrate the performance differences of communication systems at different transmission rates? I don't see a significant performance difference. I think that when the transmission rate changes, the quality of the received signal will also change. Can you explain the difference between the two different rates in Figure 8.

Answer: Thank you for your question. Figure 8 has a picture put duplicated, both are 10Mbps, here should be a 5Mbps eye diagram and a 10Mbps eye diagram, these two diagrams mainly illustrate the transmitter transmits the modulated LED light signal and the receiver receives the light signal, and observes its eye diagram with an oscilloscope to see the good characteristics. When the transmission rate changes the received signal quality will definitely change, in this circuit design, the signal quality of 5Mbps transmission is a little better than 10Mbps transmission. Now replace the other pictures to expand the illustration.(Modified in the paper)

  1. The modulation mode used for the data in Figure 8 is not explained, including the modulation mode used by the underwater communication system finally.

Answer: This paper focuses on OOK modulation using the classical Bias-T circuit model. (Modified in the paper). The original text is modified (first paragraph of Section IV) to read: Due to the complexity of the underwater environment, in order to ensure that the transmission distance of the underwater optical communication system is long, the transmitter adopts the OOK modulation technology with the Bias-T circuit model structure, and designs a high-power LED driver circuit to make the LED transmitter optical power reach 5W while the modulation rate of the LEDs can reach 10 Mbps.Principle of the high-power LED modulation technology The block diagram is shown in Figure 4-1. High-power LED modulation technology consists of five modules: level conversion, GAN driver, high-speed switching, DC bias, feedback control and monitoring. (Modified in the paper)

  1. From the perspective of communication common sense, it is difficult to avoid the bit error in the actual digital communication system. However, as can be seen from the data in Table 2, the BER from 15m to 30m is 0. Can this BER data be understood as the data after error correction? Whether the BER without error-correcting coding has been tested?

Answer: Thank you for your question. This system does not use error correction coding techniques and only BER tests were done.

  1. Finally, I suggest that the models and manufacturers of LED and PD devices used in this paper should be indicated, and all simulation software and methods should also be listed. The adjustment range of automatic gain control used in the system and the saturation problem of PMT can also be explained.

Answer: LED Model: Lattice 45mil 450nm Hangzhou Yuanfang Optoelectronic Information Co.

  SA 38mil 525nm Hangzhou Yuanfang Optoelectronic Information Co. 

PD Model: H14447 Hamamatsu Photonics Japan, Inc.

Software: MATLAB.

Automatic gain control mainly utilizes a detector circuit to detect the peak-to-peak value of the signal and then dynamically adjusts the amplifier's gain change.

PMT belongs to highly sensitive photodetectors with a large internal gain. When the light is too strong, the PMT is saturated, resulting in the PMT losing the ability to work, and if the PMT is saturated for a long period of time, the PMT will be permanently damaged or burned out.

Reviewer 2 Report

Comments and Suggestions for Authors

In this paper, a 120° large divergence angle and wavelength division duplex underwater optical communication system is designed with 10-LED array light source and PMT photodetector, and the 60° receiving field of view angle of the receiving end, which is beneficial for the optical communication of underwater mobile platform. The experimental test of the system in the swimming pool shows that it can realize the transmission distance of 30 meters, the rate of 10Mbps and low bit-error-rate(BER) communication, with the attenuation coefficient of water of 0.8dB/m, and supports the transmission of data, images and real-time video. This work is of practical significance. However, this paper has many shortcomings as follows.

1. In the introduction of the designed system in Fig. 1, it is mentioned that the system also includes “signal processing FPGA and computer control software”. The work of FPGA is very important, but it is not introduced in detail in the paper. Why? In addition, the “computer control software” is not introduced as well. Is it used on the PC, or used elsewhere? please elaborate it.

2. Why is the symbol before PR in Eq. (1) "+" instead of "-"? Is there a problem with the expression of the denominators of Eq. (2) and Eq. (3)? What the variables g and j stand for respectively is not explained. In addition, the link distance marked in the figure is lowercase "d", is it the same as the capital letter "D" in Eq. (2) and Eq. (3)?

3. What is the MCU model in the schematic circuit diagram in Fig.6? What does it do? What is the range of analog and digital quantities of its specific inputs and outputs? What kind of circuit is the "plastic circuit" in the second box in Fig. 7?

4. The paper is mainly about engineering application, and the introduction of work innovation is not enough. In particular, what are the new scientific and technical problems encountered in installing the system on an underwater mobile robot platform and using in dynamic underwater environments? How to solve these problems, please give a detailed analysis and explanation. It should not only increase the beam divergence angle to solve all the problems involved in mobile platform optical communication.

5. The output of the light source is a Gaussian beam? Or what kind of beam? Since it involves calculating the geometric loss of optical signal and the margin of link power budget please give more practical considerations and calculations.

6. The expression and grammar of this paper is required to be further improved and polished. And please pay attention to the capital and small letters in units, for example the dBm is written in the dbm in multiple places, and so on. The multiplicative sign in the text is represented by "*", which is easy to be confused as a convolution symbol, etc. Please change to the commonly used multiplicative sign.

7. The writing of the references after the text is not standardized.

Comments on the Quality of English Language

The expression and grammar of this paper is required to be further improved and polished.  

Author Response

Responses to Reviewer's Comments (Reviewer 2)
Author's Notes
Thanks for your suggestion, I have revised the article as follows:

  1. In the introduction of the designed system in Fig. 1, it is mentioned that the system also includes “signal processing FPGA and computer control software”. The work of FPGA is very important, but it is not introduced in detail in the paper. Why? Ln addition, the “computer control software" is not introduced as well. ls it used on the PC, or used elsewhere? please elaborate it.

Answer: Thank you for your question. This paper focuses on the design of large dispersion angle as well as high power LED modulation technology, large field of view reception angle design and high sensitivity detection technology.The main function of FPGA is that the information processing unit FPGA mainly completes the reception of the data stream sent by the data source through the network communication method, and processes the data with 8B/10B encoding, parallel-serial conversion, etc.; and transmits the signal to the LED driving circuit with LVTTL level. The FPGA at the receiving end performs serial-to-parallel conversion and 8B/10B decoding of the received LVTTL level; and returns the information to the terminal at the receiving end through network communication. (Modified in the paper)

  1. Why is the symbol before PR in Eq. (1) "+" instead of "_"?ls there a problem with the expression of the denominators of Eq(2) and Eq. (3)? What the variables g and j stand for respectively is not explained. ln addition, the link distance marked in the figure is lowercase "d", is it the same as the capital letter "p" in Eq. (2) and Eq.(3)?

Answer: PR represents the sensitivity of the photodetector, and the sensitivity is expressed as a "-" value, with two negative signs superimposed to make a positive sign. Equations 2 and 3 do not appear in the g, j variables, the link distance labeled in the figure is lowercase "d", whether with formula (2) and formula (3) in the capital letter "D" is the same.(Modified in the paper)

  1. What is the MCU model in the schematic circuit diagra min Fig.6? What does it do? What is the range of analog and digital quantities of its specific inputs and outputs? What kind of circuit is the "plastic circuit" in the second box in Fig. 7?

Answer: Thank you for your question. MCU is used is a microcontroller C8051F007 as the control core, here is a circuit model diagram does not take into account the specific parameters of the design, the input and output analog and digital and GANFET tube sampling resistance.The second box "plastic circuit" in Fig. 7 is to indicate the level conversion circuit, because the automatic gain control circuit comes out with non-standard LVTTL level signals, so it is necessary to add a level conversion circuit to uniformly convert the signal level to LVTTL level signals and input them into the FPGA for judgment. (Modified in the paper)

  1. The paper is mainly about engineering application, and the introduction of work innovation is not enough. In particular, what are the new scientific and technical problems encountered in installing the system on an underwater mobile robot platform and using in dynamic underwater environments? How to solve these problems, please give a detailed analysis and explanation. Lt should not only increase the beam divergence angle to solve all the problems involved in mobile platform optical communication.

Answer: That's a good comment. This paper indeed focuses on engineering applications, mainly introducing four aspects of large divergence angle design, high power LED modulation technology, large field of view reception technology, and high sensitivity detection technology.Mainly to solve the underwater robot in the process of movement may appear jitter, as well as the two are not in the same level, underwater alignment difficulties and other practical problems, the use of these four technologies can be realized in the synthesis of a more stable underwater mobile inter-robot communication.

  1. The output of the light source is a Gaussian beam? Or what kind of beam? Since it involves calculating the geometric loss of optical signal and the margin of link power budget please give more practical considerations and calculations.

Answer: The output of the light source is a Gaussian beam, due to the complexity of the underwater environment, the link budget has been made to consider the impact of a variety of attenuation characteristics, while also introduced in the article in order to cope with the complexity of the underwater environment, the redundancy of the communication link is often greater than 5dbm in order to ensure the reliability of the communication.

  1. The expression and grammar of this paper is required to be further improved and polished. And please pay attention to the capital and small letters in units, for example the dBm is written in the dbm in multiple places, and so on. The multiplicative sign in the text is represented by "*", which is easy to be confused as a convolution symbol, etc. Please change to the commonly used multiplicative sign.

Answer: Thank you for your comment. The paper has been modified.

  1. The writing of the references after the text is not standardized.

Answer: Thank you for your comment. Modifications were made according to the norms of the reference literature.

Reviewer 3 Report

Comments and Suggestions for Authors

The article mainly studies the optical communication between underwater mobile robots based on blue-green LED. Overall it has practical significance but still has some places for improvement. Here below are some detailed suggestions and questions.

1.       on page 2, What is “LVTTL”?

2.       On page 3 and page 4, What is the difference between “PLm” in Eq.(1) and “PLM” in Eq.(3)?

3.       What is the difference between “D” in Table 1 and “d” in Figure 2?

4.       On page 3, what is “φ” in Eq.(2)?

5.       On page 4, In Figure 3, what does the “PLM” of the Y-axis represent? Is it PLM or PLM?

6.       What software was used for the simulation in the experiment? For example, Figure 4 and Figure 5.

7.       On page 5, in Eq.(4), please give the definition ofω.

8.       In Figure 5, the graphics are unclear and the resolution is low. Please optimize the graphics.

9.       For Figure 8, please provide a detailed explanation of the differences between the left and right graphs in Fig. 8(a) and 8(b).

10.    Some of the words are just mentioned in abbreviated form in the text (e.g.ROVs, TIA, MCU, UART (Fig.6), OPA(Fig.7), AWG, PM, LD, OS, PC in Fig. 3, etc.). Please indicate the full name of the abbreviation in their first appearance in the text and use the abbreviated form thereafter.

11.    For PMT, it is already spelled out completely when it first appears on page 2, line 9, and there is no need to explain it repeatedly in subsequent sections, for example on page 2, line 12 from the bottom; on page 6, line 3 from the bottom;

12.    According to the section of “5. Underwater Optical Communication Experiment,” “…the high-definition video, high-definition image and other data types.” in the “Abstract” section have not been studied, please modify the description.

13.    The authors should correct the References section. The list of references is not in the Photonics style.

Comments on the Quality of English Language

Moderate editing of English language required

Author Response

Responses to Reviewer's Comments (Reviewer 3)
Author's Notes
Thanks for your suggestion, I have revised the article as follows:

  1. On page 2, What is “LVTTL”?

Answer: LVTLL is a level logic signal. The provision reads as follows:3.3V LVTTL: Vcc: 3.3V, VOH>=2.4V, VOL<=0.4V, VIH>=2V, VIL<=0.8V.

  1. On page 3 and page 4, What is the difference between“PLm” in Eq.(1) and “PLM" in Eq.(3)?

Answer: Thank you for your question. This stands for the same meaning, which has been harmonized and modified to PLM.

(Modified in the paper)

  1. What is the difference between “D” in Table 1 and “d” in Figure 2?

Answer: Both express the underwater communication distance, which has been standardized as d.(Modified in the paper)

  1. On page 3, what is “φ" in Eq.(2)?

Answer: φ is an over-typed letter and here it should be a θ.(Modified in the paper)

  1. On page 4, In Figure 3, what does the “PLM” of the Y axis represent? ls it PLM or PLM?

Answer: PLM indicates the amount of link redundancy that is PLM.(Modified in the paper)

  1. What software was used for the simulation in the experiment? For example, Figure 4 and Figure 5.

Answer: Simulated with Ligthtools.(Modified in the paper)

  1. On page 5, in Eq.(4), please give the definition of w.

Answer: w denotes the receiving field-of-view half-angle.(Modified in the paper)

  1. In Figure 5, the graphics are unclear and the resolution is low. Please optimize the graphics.

Answer: Changes have been substituted in the paper.

  1. For Figure 8, please provide a detailed explanation of the differences between the left and right graphs in Fig. 8(a)and 8(b).

Answer: The graph here puts up two identical graphs, which should be a 5Mbps graph and a 10Mbps graph.(Modified in the paper)

  1. Some of the words are just mentioned in abbreviated form in the text (e.g.ROVs, TIA, MCU,UART (Fig.6),OPA(Fig.7), AWG, PM, LD, OS, PC in Fig. 3, etc.). Please indicate the full name of the abbreviation in their first appearance in the text and use the abbreviated form thereafter.

Answer: All the full name has been modified in the paper.

  1. For PMT, it is already spelled out completely when it first appears on page 2, line 9, and there is no need to explain it repeatedly in subsequent sections, for example on page2, line 12 from the bottom; on page 6, line 3 from the bottom;

Answer: Thank you for your comment. The paper has been modified.

  1. According to the section of “5. Underwater Optical Communication Experiment,” “...the high-definition video high-definition image and other data types." in the“Abstract section have not been studied, please modify the description.

Answer: Thank you for your comment. Underwater robots have been widely used in the fields of ocean exploration, seabed operation, and marine scientific research. And the underwater low latency, efficient and safe communication method is the key to realize the applied underwater robot data transmission system. This paper focuses on the optical communication between underwater mobile robots including large divergence angle LED design, large field-of-view receiving technology, and weak light detector technology.Under the condition that the LED transmitting angle is 120°, the receiving field of view angle of the system reaches 60°, and after experimental testing, the mobile communication between underwater mobile robots can stably transmit the information at a communication distance of 30 meters and the communication rate is maintained at more than 10Mbps. This study provides a solution for cable-free data transmission for underwater robots which can improve the application areas of underwater robots. (Modified in the paper)

  1. The authors should correct the References section. The list of references is not in the Photonics style.

Answer: Thank you for your comment. The paper has been modified.

Round 2

Reviewer 1 Report

Comments and Suggestions for Authors

Thank you for answering my questions one by one, you have done a lot to improve.But there are still some problems that need to be improved. 

1.     The picture quality of Fig. 4 and Fig. 5 is still low, and it still feels like a screenshot. Can the simulation software export high-definition vector images? The two figures in Fig. 4(b) and Fig. 5(b) should be the important conclusions of the simulation, but the information and text in the figures are very small and almost invisible. It is suggested to redraw the coordinate axes and data information.

2.      Should there be a corresponding reference legend for the light intensity distribution diagram in Figure 5(b)? (To tell people what is the corresponding light intensity of different colors.)

3.      Now that you have the actual system, could you please add some measured data? Most of the data in this paper, such as light intensity distribution at different angles between the transmitting end and the receiving end and the attenuation degree of light to the water body, are simulation results. If conditions permit, one or two sets of measured data can be added, so that the paper can combine theory and practice, and the workload will be fuller.

Reviewer 2 Report

Comments and Suggestions for Authors

1. In this paper, only one optical communication experiment through their designed system is conducted on the mobile platform of underwater robot, and the measure to cope with the platform movement is only to increase the divergence angle of the emitted light. However, there is no in-depth study on how to ensure the communication quality of the designed optical communication system under various mobile states of the platform. The 10Mbps rate and 30m transmission distance achieved in this paper, and even the light divergence angle is increased, similar results have been published in several papers. Therefore, the innovation and contribution of this paper are very limited, and this paper is not suitable for publication in Photonics at present.

2. The FPGA is used in the design of optical transmitter and optical receiver, but the paper does not elaborate the working process and working sequence diagram of FPGA in detail.

3. Is Figure 8 an eye diagram of the OOK modulated signal? What instrument is used to obtain it? Please give the specific instrument model. What are the bit error rates of 5Mbps and 10Mbps signals respectively? In addition, these two figures are still unclear in the revised manuscript.

4. The BERs of the transmitting distances of 15m to 30m are all 0, how is it measured? The variation curve of BER with transmission distance is not seen in the paper.

5. Regarding the design of automatic gain control circuit, please give the parameters of the specific circuit and the dynamic range of the optical receiver.

6. Due to the high sensitivity of PMT, the reception of large signals is easy to saturate, and even be burned out. Has the protection circuit been designed in the receiver?

7. In addition, the revised manuscript did not give further in-depth study, revision and answer to according to the comments made by the three reviewers of the first round review.

Comments on the Quality of English Language

The English expression about this paper needs to be polished further.

Reviewer 3 Report

Comments and Suggestions for Authors

The data Figures are still unclear. Especially, Figures 4 and 5. For except, Figures 4,5 and 8 look like screenshots, which need to be modified for academic articles.

Round 3

Reviewer 2 Report

Comments and Suggestions for Authors

The new manuscript has been revised one by one in response to the problems raised by the reviewer, and it is suggested that it be accepted and published after minor revisions.

(1) Please modify the format of diagrams and pictures, tables and references, according to the requirements of Photonics.

(2) The English expression of the manuscript needs further to be polished.
